# WICE: Real-World Entailment for Claims in Wikipedia

**Ryo Kamoi**◇    **Tanya Goyal**◇    **Juan Diego Rodriguez**♠◇    **Greg Durrett**◇

◇ Department of Computer Science, The University of Texas at Austin
♠ Applied Research Laboratories, The University of Texas at Austin
ryokamoi@utexas.edu

## Abstract

Textual entailment models are increasingly applied in settings like fact-checking, presupposition verification in question answering, or summary evaluation. However, these represent a significant domain shift from existing entailment datasets, and models underperform as a result. We propose WICE, a new fine-grained textual entailment dataset built on natural claim and evidence pairs extracted from Wikipedia. In addition to standard claim-level entailment, WICE provides entailment judgments over subsentence units of the claim, and a minimal subset of evidence sentences that support each subclaim. To support this, we propose an automatic claim decomposition strategy using GPT-3.5 which we show is also effective at improving entailment models' performance on multiple datasets at test time. Finally, we show that real claims in our dataset involve challenging verification and retrieval problems that existing models fail to address.[1]

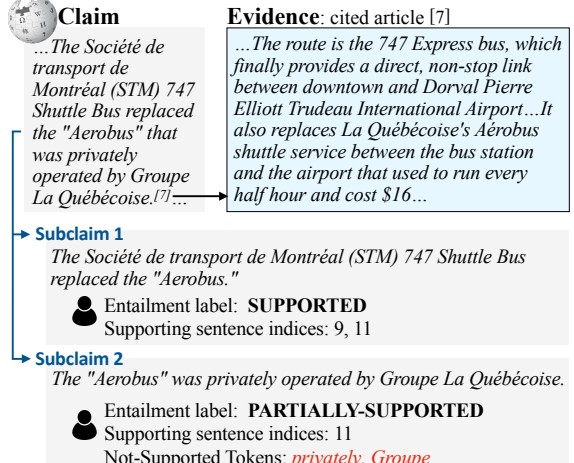

**Claim**
*...The Société de transport de Montréal (STM) 747 Shuttle Bus replaced the "Aerobus" that was privately operated by Groupe La Québécoise.[7]...*

**Evidence**: cited article [7]
*...The route is the 747 Express bus, which finally provides a direct, non-stop link between downtown and Dorval Pierre Elliott Trudeau International Airport ...It also replaces La Québécoise's Aérobus shuttle service between the bus station and the airport that used to run every half hour and cost $16...*

→ **Subclaim 1**
*The Société de transport de Montréal (STM) 747 Shuttle Bus replaced the "Aerobus."*

👤 Entailment label: **SUPPORTED**
Supporting sentence indices: 9, 11

→ **Subclaim 2**
*The "Aerobus" was privately operated by Groupe La Québécoise.*

👤 Entailment label: **PARTIALLY-SUPPORTED**
Supporting sentence indices: 11
Not-Supported Tokens: *privately*, *Groupe*

Figure 1: WICE annotation for a claim in Wikipedia and its cited evidence. Claims are automatically broken into subclaims. WICE provides entailment labels and indices of evidence sentences that support a subclaim. Real-world claims are often partially supported (subclaim 2); unsupported tokens are annotated here.

## 1 Introduction

Textual entailment (Dagan et al., 2005) and natural language inference (MacCartney and Manning, 2009; Bowman et al., 2015; Williams et al., 2018) are longstanding problems in NLP that take many forms. The SNLI dataset has a stated purpose to use NLI "*as a tool for the evaluation of domain-general approaches to semantic representation*" (Bowman et al., 2015). However, this is far from how NLI is used today, e.g., to validate QA system outputs (Chen et al., 2021), evaluate generated summaries (Falke et al., 2019; Laban et al., 2022) or understand knowledge-grounded dialog (Honovich et al., 2021; Gupta et al., 2022; Dziri et al., 2022).

There are some major gaps when applying modern entailment systems to these tasks. First is the fact that many NLI datasets target short premises, often single sentences, such as VitaminC (Schuster et al., 2021) and WANLI (Liu et al., 2022). As a result, existing frameworks for document-level entailment are built upon aggregating local entailment scores (Zhou et al., 2019; Laban et al., 2022) or using a retrieval stage (Nie et al., 2019; Schuster et al., 2022). There are a few exceptions such as DocNLI (Yin et al., 2021), but it features a large amount of synthetic negative data. This highlights the second weakness: the lack of ecologically valid negative examples. The process by which contradictory cases are authored leads to spurious correlations, including single-word correlations (Gururangan et al., 2018; Gardner et al., 2021), syntactic heuristics (McCoy et al., 2019), or a lack of reliance on the input (Poliak et al., 2018). Third, these datasets lack fine-grained annotations of what parts of a claim are supported or not.

Our work addresses these shortcomings by collecting WICE (Wikipedia Citation Entailment), a dataset for verification of real-world claims in

---

[1]Our data is available at: https://github.com/ryokamoi/wice

Wikipedia. Given a sentence in Wikipedia and the corresponding article(s) it cites, we annotate the entailment label, a list of sentences in the cited article(s) that support the claim sentence, and tokens in the claim that are unsupported by the article(s) (see Figure 1). We show that the claims in WICE involve challenging verification and retrieval problems beyond the scope of current NLI datasets.

To aid the construction of WICE and provide fine-grained annotations, we introduce Claim-Split, a method of decomposing hypotheses into subclaims using few-shot prompting with GPT-3.5[2] (Brown et al., 2020; Ouyang et al., 2022), shown in Figure 2. This decomposition resembles past frameworks derived from OpenIE (Stanovsky et al., 2018; Ernst et al., 2021) or Pyramid (Nenkova and Passonneau, 2004; Shapira et al., 2019; Zhang and Bansal, 2021), but avoids relying on annotated data and achieves greater flexibility by using GPT-3.5. By operating at the subclaim level, we simplify both our annotation process and the final entailment prediction task for automatic models. We also show that Claim-Split can improve the entailment classification performance of off-the-shelf models by simplifying long claims.

We evaluate a range of systems on our dataset, including existing short-paragraph entailment models "stretched" to make a document-level entailment judgment out of short-paragraph judgments (Laban et al., 2022; Schuster et al., 2022). We show that chunk-level processing of the long evidence and retrieval-based approach are a strong starting point for future systems, although current systems still perform below human level on this dataset and retrieval performance is poor.

## 2 The WICE Dataset

We aim to annotate claims that are: (1) ***naturally-occurring***; we extract claims from Wikipedia and its cited documents, where the noise in citations gives realistic negative examples, (2) ***in-context*** with surrounding text; this mirrors real use cases where claims occur in discourse context, and (3) ***fine-grained***; we break down complex claims into multiple subclaims with Claim-Split and provide entailment judgments for both granularities. We also provide token-level annotation for non-supported tokens.

---

[2]GPT-3.5 was the strongest model available at the time we collected the data.

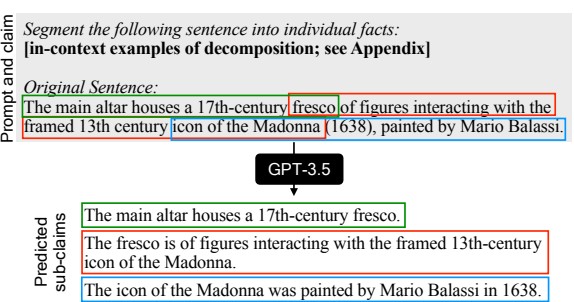

Figure 2: Claim-Split automatically breaks claims into subclaims using a LLM (GPT-3.5 in this work).

### 2.1 Claim-Split using LLMs

A key idea we argue for in this paper is **claim decomposition**. Real world claims, such are those in Wikipedia that constitute WICE, often consist of multiple related pieces of information, each of which may or may not be supported by the evidence. We show an example of this in Figure 1 where the two subclaims, derived from the complex claim, have different entailment labels. By decomposing claims prior to annotation, and collecting annotations at the subclaim level, we can offer a more fine-grained view into which parts of claims are supported by cited documents.

First, we describe our claim decomposition strategy, called Claim-Split. Then, we establish the validity of Claim-Split as a pre-annotation step by manually verifying the generated subclaims.

**Claim-Split method** Our method prompts an instruction-tuned GPT-3.5 model (text-davinci-002) (Brown et al., 2020; Ouyang et al., 2022) in a few-shot setting to automatically decompose a claim $c$ into multiple subclaims: Claim-Split$(c) = \{c_1, ...c_m\}$. The prompt used includes $K$ example splits along with the instruction "*Segment the following sentence into individual facts.*" These examples are designed such that the subclaims provide complete coverage over the corresponding input claims (see prompt used for WICE in Appendix D.1). Figure 2 shows an example of decomposed claims using our approach.

**Claim-Split generated subclaims correctly and subclaims completely cover the information in the source claim.** We recruited Mechanical Turk workers to annotate subclaims decomposed by Claim-Split for 700 Wikipedia claims,[3] for two necessary criteria: (1) **Completeness**: subclaims

---

[3]We explain how claim sentences are chosen in Section 2.3.

$\{c_1, \ldots c_m\}$ must cover all information in $c$, and (2) **Correctness**: all subclaims $c_i$ must faithfully present a part of the information in $c$.

We found that $92.3\%$ of the claims satisfy the completeness criterion and $97.7\%$ of the generated subclaims satisfy the correctness criterion. Analyzing the errors further, we found that one-third were relatively trivial such as disregarding parentheses in the original claim, and could be solved with targeted prompts. Other errors were more complex and do not have straightforward solutions; see Appendix A for more analysis. In total, only $8.6\%$ of the claims included one of the two types of errors. **These results justify our strategy of seeking annotations at the subclaim level** and show that building systems at the subclaim level is a viable path for further efforts.

The set of annotated claims from this experiment constitutes the complete dev and test set of WICE (details later in Section 2.3); therefore, we manually fix the errors detected in this annotation to provide a high quality evaluation dataset.

## 2.2 Tasks in WICE

Let claim $c$ (analogous to the hypothesis in standard NLI terminology) be a sentence in a Wikipedia article, and evidence $E = \{e_1, \ldots, e_n\}$ (analogous to the premise) refer to sentences from web article(s) cited as a reference for the claim $c$. Let `Claim-Split`$(c) = \{c_1, \ldots, c_m\}$ be the automatically decomposed subclaims from $c$.

The human-annotated data we collect supports the following three tasks:

1. **Entailment Classification**: Given a claim $c$ (or subclaim $c_j$) and evidence document $E$, is the claim (or subclaim) entailed by the document? We annotate three-way entailment: {SUPPORTED, PARTIALLY-SUPPORTED, NOT-SUPPORTED}.

2. **Evidence Retrieval**: Given a claim $c$ (or subclaim $c_j$) and evidence document $E$, which subset of evidence sentences $\{e_1, \ldots, e_k\} \subset E$ support or partially support $c$ (or $c_j$)?

3. **Detecting Non-Supported Tokens**: Given subclaim $c_j$ and evidence document $E$, which tokens $\{t_1, \ldots, t_p\} \subset c_j$ are not supported by $E$?

For each of these three sub-tasks, we only collect human annotations at the subclaim-level (as shown in Figure 1). Claim-level labels are obtained by automatically projecting subclaim annotations using a natural set of rules described in Section 2.3.

## 2.3 Dataset Collection

**Base Data** We use the same base set of Wikipedia claims as the SIDE dataset (Petroni et al., 2023). For each claim, we re-retrieve the cited web article(s) from Common Crawl[4] and segment into sentences.[5] This gives us our base set of claim-evidence pairs $(c, \{e_1, e_2, \ldots e_n\})$.

**Claim-Split** Next, we split each claim $c$ into subclaims $\{c_1, c_2, \ldots c_m\}$ using `Claim-Split`, as described in Section 2.1. We use few-shot prompting with six examples (Appendix D.1). Also, we filter claims that are decomposed into either only one or more than six subclaims.[6] For the development set, this filters $19.1\%$ of examples.

**Additional Filtering** We use a NLI model (`RoBERTa-Large`) trained on DocNLI (Yin et al., 2021) to remove datapoints $(c, E)$ for which **all** subclaims $c_j \in$ `Claim-Split`$(c)$ are classified as *entailed*.[7] By using a relatively weaker `RoBERTa-Large` model, we remove trivially entailed claims but avoid making a dataset that is adversarially difficult for larger models (e.g., T5-3B).

We retain $16.3\%$ of the claims after applying all the filtering steps above. Despite this, we observe diverse entailment phenomena in the remaining subset, which we analyze further in Section 2.5.

**Human Annotation** We recruited Mechanical Turk workers to annotate evidence-subclaim pairs for each of the three tasks outlined in Section 2.2. First, we ran a qualification task with 3 examples (chosen to include challenging annotations) and qualified 23 workers based on it. Annotators were paid $0.75 per HIT. Each HIT involved annotation of all subclaims corresponding to a single claim.

We collect annotations from 5 unique workers for each example in the development and test set, and from 3 for the train set. We observe reasonable inter-annotator agreement for the entailment

---

[4] https://CommonCrawl.org/.

[5] We re-retrieved the cited web articles because SIDE only contains one evidence web article even when there are multiple on the original Wikipedia page.

[6] We removed the claims that cannot be decomposed by `Claim-Split` because we intend to focus on complex claims. In addition, we removed the claims that are decomposed into more than six sub-claims because typically they are not interesting cases; they often just include a list of examples.

[7] Detail of how we split $E$ into chunks and combine chunk-subclaim entailment scores are in Appendix B.

| Statistic | Subclaim | Claim |
|---|---|---|
| # Datapoints – Train | 3,470 | 1,260 |
| Dev | 949 | 349 |
| Test | 958 | 358 |
| # Tokens | 12.1 | 27.4 |
| # Supporting Sents | 1.9 | 3.1 |
| # Subclaims / Claim | – | 3.0 |
| # Evidence Sentences / Datapoint | 119.5 | |
| # Tokens / Evidence Sentence | 14.0 | |
| # Tokens / Claim's Context | 122.5 | |

Table 1: Statistics of the WICE dataset.

| | Supported | Partially Supp. | Not Supp. |
|---|---|---|---|
| **Claim** | 33.0 | 54.7 | 12.3 |
| **Subclaim** | 55.8 | 18.2 | 25.9 |

Table 2: Entailment label distribution (%) of claims and subclaims in the development set of WICE.

| Category | FEVER | VitC | WICE Subcl | WICE Claim |
|---|---|---|---|---|
| **Compression** | 10 | 20 | 3.9 | 4 |
| w/ contextualization | 28 | 14 | 14.2 | 4 |
| **Paraphrase**   Direct | 34 | 24 | 26.0 | 16 |
| + Calculation | 0 | 30 | 0.0 | 0 |
| + Inference | 24 | 8 | 52.8 | 68 |
| + Background Knowledge | 2 | 0 | 3.1 | 8 |
| Annotation Mistake | 2 | 4 | 0.0 | 0 |

Table 3: Distribution (%) of verification problems estimated from annotation for 50 claims in each dataset (127 subclaims in WICE). We evaluated claims labeled as entailed in FEVER and VitaminC, and claims labeled as supported or partially supported in WICE.

classification task; Krippendorff's $\alpha = 0.62$ on the development set. We describe how we aggregate annotations from these workers in Appendix B.

**Deriving claim labels from subclaim labels** For *entailment classification*, if all subclaims are SUPPORTED or NOT-SUPPORTED, we assign that as the claim-level label. Else, we assign PARTIALLY-SUPPORTED. For *supporting sentences*, we take the union of subclaim level supporting sentences as the claim level annotation. When there are multiple sets of supporting sentences for each subclaim,[8] we take the union of all combinations.

## 2.4 Dataset Statistics

Table 1 shows the overall statistics for the final WICE dataset. On average, claims were decomposed into 3.0 subclaims by the Claim-Split method. Our dataset contains approximately 5.9K subclaim level (or 2K claim level) examples for the entailment classification task. Each subclaim (or claim) is supported by an average of 1.9 (or 3.1) evidence sentences when the label is SUPPORTED or PARTIALLY-SUPPORTED.

Table 2 shows the distribution of entailment labels. Roughly 56% of the subclaims are labeled as SUPPORTED. This percentage is much lower at the claim level (33%) since *all* subclaims must be SUPPORTED. Consequently, a majority of the claims fall into the PARTIALLY-SUPPORTED category. In a

typical NLI dataset, both PARTIALLY-SUPPORTED and NOT-SUPPORTED would simply be labeled as neutral, which is not very useful for a system designer attempting to verify a fact. For PARTIALLY-SUPPORTED subclaims, 25.2% of tokens were identified as not supported.

## 2.5 Analysis of Phenomena

We compare with FEVER and VitaminC to show that the in-the-wild claims and cited articles in WICE constitute diverse and challenging verification problems. We bucket verification problems from these datasets into the following categories. First, we define **Compression** to include cases where the claim/subclaim appears almost verbatim in the evidence document with trivial phrasal deletions. Relatedly, the category **Compression w/ Decontextualization** includes cases in which pronouns, VP ellipsis, or other similar effects need to be resolved to appropriately contextualize the claim. These are typically not challenging for language models. We also define four cases we broadly call **Paraphrasing**. Within this, **Direct** cases are those where the evidence document information has been restated in the claim, but in a way that is relatively transparent and simple to follow (e.g., synonym substitution). **Require Calculation** includes cases that require numerical calculation or comparison. **Require Inference** captures slightly more complicated where inferences about the situation are needed. **Require Background Knowledge** represents cases where additional background knowledge or world knowledge is required (e.g., knowledge of entity aliases or the ability to recognize causal relationships between events).

We manually annotate 50 claims randomly se-

---

[8]There can be multiple sets of supporting sentences for each claim/subclaim because different annotators can annotate different sets of supporting sentences that include sufficient information to support (or partially support) the claim/subclaim.

| Model | Train Data | Claim | | Subclaim | |
|---|---|---|---|---|---|
| | | F1 | Acc | F1 | Acc |
| Majority Label | – | 49.6 | 33.0 | 71.6 | 55.8 |
| **Sentence-level** | | | | | |
| RoBERTa-Large | SNLI[1] | 47.3[†] | 31.0[†] | 73.3[†] | 66.8[†] |
| RoBERTa-Large | MNLI[1] | 46.5[†] | 35.8[†] | 71.9[†] | 57.9[†] |
| ALBERT-xLarge | VitaminC[2] | 49.8[†] | 58.9[†] | 74.1[†] | 67.8[†] |
| ALBERT-xLarge | VitC+MNLI[2] | 52.7[†] | 61.5[†] | 77.0[†] | 73.0[†] |
| T5-3B | VitC+MNLI | 48.3[†] | 61.7[†] | 77.1[†] | 73.4[†] |
| T5-3B | ANLI | 48.0[†] | 41.9[†] | 77.6[†] | 71.8[†] |
| **Chunk-level** | | | | | |
| T5-3B | DocNLI | 56.4[†] | 62.8[†] | 79.7[†] | 74.9[†] |
| T5-Large | ANLI | 61.8 | 70.7[†] | 80.7 | 78.3 |
| T5-3B | ANLI | **64.3** | **75.1** | **83.4** | **79.6** |
| Human | – | 83.3 | 92.0 | 94.4 | 94.4 |

Table 4: **Off-the-shelf** binary entailment classification performance of existing NLI models on WICE using the MAX strategy to combine local entailment scores. [1]: Dataset-model configurations from Laban et al. (2022) and [2]: from Schuster et al. (2021). Similar to Honovich et al. (2021), we observe the best performance using T5 models trained on ANLI, although there remains a gap between these and human performance. [†]: Worse than T5-3B trained on ANLI (chunk-level) with p-value $< 0.05$ according to a paired bootstrap test.

lected from the development sets with these categories for the three datasets.[9] In WICE, we annotated 127 subclaims in 50 claims. Table 3 shows the estimated distribution. We can see that **natural claims in WICE involve difficult entailment classification problems often requiring some kind of inference even at the subclaim level**. In contrast, relatively few claims in VitaminC involve inference, but mostly require narrower types of reasoning such as calculation.

## 3 Experiments on WICE

We have three main questions for our dataset. (1) How well do existing NLI models perform off-the-shelf when using the "stretching" paradigm? (2) Does fine-tuning on our dataset improve accuracy? (3) Would being able to retrieve the relevant supporting sentences improve accuracy further?

### 3.1 Entailment Classification on WICE

We benchmark the performance of NLI models on WICE in both off-the-shelf and fine-tuned settings.

[9]If more than one category applies, we assign the most "difficult" category (latest in our list). Examples are given in Table 13 in the appendix.

**Stretching NLI for document-level entailment** WICE's evidence articles are generally much longer than the input length limits of NLI models. Therefore, we adopt the "stretching" technique from prior work (Laban et al., 2022; Schuster et al., 2022) for all our entailment models.

We divide the evidence document $E$ into multiple partitions $P_E$. These can be individual sentences ($e_i$) or chunks (contiguous $e_i$'s). We restrict maximum chunk size to 256 tokens. For each subclaim/claim and partition pair, we use the NLI model to get a "local" entailment score: $sc(c_i, p) = P(y = \text{entailed} \mid c_i, p)$. Then, as the first method of the "stretching" technique, we derive a document-level score by taking the maximum local score: $sc(c_i, E) = \max_{p \in P_E} [sc(c_i, p)]$. We call this the MAX entailment strategy.

**Off-the-shelf models** We report the performance of RoBERTa-Large (Liu et al., 2019), ALBERT-xLarge (Lan et al., 2020), T5-Large and T5-3B (Raffel et al., 2020) fine-tuned on SNLI (Bowman et al., 2015), MNLI (Williams et al., 2018), DocNLI (Yin et al., 2021) or ANLI (Nie et al., 2020). We choose model-dataset pairs that have been used in prior work and have released weights, but default to T5-3B for our new settings.

The PARTIALLY- and NOT-SUPPORTED labels in WICE correspond to the neutral (and sometimes contradiction) category in SNLI, MNLI, and ANLI. On the other hand, DocNLI only includes binary categories (entailed or not entailed). To evaluate models trained on these datasets on WICE, we consider a binary classification task: SUPPORTED or not. We use the predicted probability for entailment as the predicted score for the SUPPORTED class.

We evaluate GPT-3.5 and GPT-4 in Section 3.4 on the oracle retrieval dataset (explained later); the stretching approach requires invoking the model for every (document chunk, subclaim) pair, which becomes very expensive to test with large models.

**Models fine-tuned on WICE** WICE consists of entailment labels corresponding to entire evidence documents and also supporting sentences for SUPPORTED and PARTIALLY-SUPPORTED cases. To train models that can be stretched as described above, we derive sentence- and chunk-level entailment labels from these WICE annotations (details are in Appendix E.2). Although we evaluate the performance on the binary classification task, we fine-tune models on the three-way classification

| Unit | Train Data | Claim | | Subclaim | |
|---|---|---|---|---|---|
| | | F1 | Acc | F1 | Acc |
| sent | ANLI+WiCE | 58.0† | 62.8† | 81.2† | 77.3† |
| chunk | WiCE | 65.3† | 77.1 | 85.1† | 82.7† |
| chunk | ANLI+WiCE | **72.1** | **79.1** | **87.3** | **85.0** |
| – | Human | 83.3 | 92.0 | 94.4 | 94.4 |

Table 5: Binary entailment classification on WICE with **fine-tuned** T5-3B models. Further fine-tuning on ANLI-tuned models improves performance and chunk-level outperforms sentence-level. †: Significantly worse than T5-3B fine-tuned on ANLI+WICE (chunk-level) with p-value $< 0.05$ according to a paired bootstrap test.

labels in WICE.

**Human Performance**   We manually annotate 50 randomly selected test claims and report that as human performance. Similar to crowd annotation, we annotated at the subclaim level and aggregated them to obtain claim-level judgments.

**Results**   Table 4 outlines **off-the-shelf** performance on WICE for the binary entailment classification task. It shows that **predicting scores at the chunk-level works better than sentence-level** using the MAX strategy. Overall, T5-3B trained on ANLI performs best, though it is **still substantially lower than human performance** (64.3 vs 83.3 F1 at the claim-level). This shows that the realistic claims and document-level setting of WICE differs substantially from previous NLI datasets.

For **fine-tuning**, we evaluate two settings: only fine-tuning on WICE or further fine-tuning a T5-3B model already fine-tuned on ANLI. Results are in Table 5. It shows that the chunk-level T5-3B model fine-tuned on WICE after ANLI achieves the best performance at both granularity levels. Although it improves over off-the-shelf results in Table 4, it is still substantially lower than human performance. This suggests that **WICE is challenging even for fine-tuned models.**

### 3.2   Evidence Retrieval on WICE

First, we benchmark the performance of sentence-level NLI models fine-tuned on WICE on the retrieval task: given claim/subclaim $c$, retrieve all supporting sentences from evidence $E$. Then, we evaluate if a retrieve-then-predict pipeline can improve the performance of entailment classification.

**Retrieval using NLI models**   Our strategy is as follows: derive a score for each evidence sentence and claim/subclaim pair. We use $p(\text{entailed})$

| Train Data | Claim | | | Subclaim | | |
|---|---|---|---|---|---|---|
| | F1 | P | R | F1 | P | R |
| BM25 | 15.7† | 9.0† | 98.6 | 13.2† | 7.8† | 96.7 |
| ANLI | 24.8† | 27.1† | 39.2† | 41.9† | 39.5† | 55.5† |
| WiCE | 49.0† | 48.4† | 67.0† | 49.3† | 46.3† | 67.6† |
| ANLI+WiCE | 62.0† | 61.0† | 76.9† | 58.5 | 54.7 | 79.6 |
| **w/ evidence context** | | | | | | |
| WiCE | **67.4** | **65.0** | **81.7** | **58.6** | **54.9** | 75.7 |
| ANLI+WiCE | 64.8† | 62.5 | 77.8† | 56.6 | 49.0 | **86.4** |
| Human | 90.9¶ | 92.2¶ | 92.6¶ | 91.6 | 93.2 | 92.5 |

Table 6: Performance of T5-3B on the **evidence retrieval task** of WICE. †: Worse than T5-3B finetuned on WICE (w/ context) with p-value $< 0.05$ in a paired bootstrap test. Human performance is on 50 random claims. ¶: For claim level human performance, we take the union set of retrieved sentences at subclaim level.

for prior NLI datasets and $p(\text{SUPPORTED}) + 0.5 \times p(\text{PARTIALLY-SUPPORTED})$ for WICE as retrieval scores.[10] Evidence sentences with scores larger than a threshold $\tau$ are predicted as supporting sentences.

However, we saw in Table 5 that sentence-level NLI models perform significantly worse than chunk-level models on classification, suggesting that a single sentence without context is insufficient for entailment evaluation. Therefore, we also train another sentence-level variant that includes additional **evidence context** (128 tokens) as input, in a format of "claim <SEP> evidence-context <SEP> evidence-sentence".[11]

**Metric**   We evaluate supported or partially-supported claims and subclaims, which include at least one gold supporting sentence. As there can be multiple gold sets of supporting sentences for each claim/subclaim in WICE,[12] we report the maximum F1 score over the gold sets for each claim/subclaim: $\max_i \text{F1}(\hat{S}_\tau, S_i)$ where $\hat{S}_\tau$ is the predicted set and $S_i$ is the $i$-th gold set. We choose the threshold $\tau$ that gives the best F1 score (calcu-

---

[10]The intuition behind this formula is that we want to prefer the sentences that receive high scores for the supported category, while also accepting partially supporting sentences.

[11]We did not observe performance improvement by including context for claims. This is likely because the evidence in our dataset already relates to the entity or event in the claim. Therefore, decontextualization is less critical here, but would likely be crucial if retrieving supporting documents.

[12]Different annotators may select different sets of supporting sentences that contain equally sufficient information to support (or partially support) a claim. WICE includes all of them as gold sets of supporting sentences.

| Setting | Claim | | Subclaim | |
| --- | --- | --- | --- | --- |
| | F1 | Acc | F1 | Acc |
| MAX best (Table 5) | 72.1 | 79.1 | 87.3 | 85.0 |
| top-k | 71.1 | 79.3 | 86.6[‡] | 84.2[‡] |
| top-k (w/ context) | **72.9** | **79.9** | **88.4** | **87.0** |
| oracle | 78.0 | 84.4 | 88.7 | 87.7 |
| Human | 83.3 | 92.0 | 94.4 | 94.4 |

Table 7: Binary entailment classification performance of the **retrieve-then-predict** pipeline using chunk-level T5-3B on WICE. [‡]: Worse than top-k (w/ context) with p-value $< 0.05$ according to paired bootstrap test.

| Model | Claim | | Subclaim | |
| --- | --- | --- | --- | --- |
| | F1 | Acc | F1 | Acc |
| T5-3B (ANLI) | 61.8 | 74.0 | 83.9 | 85.0 |
| T5-3B (ANLI+WICE) | 77.8 | 88.0 | 88.7 | 89.0 |
| GPT-3.5 | 39.3 | 32.0 | 73.3 | 73.3 |
| GPT-4 | 61.0 | 77.0 | 91.1 | 92.0 |
| Human | 83.3 | 92.0 | 94.4 | 94.4 |

Table 8: Binary entailment classification performance of the GPT models (few-shot) on WICE with the **oracle** retrieval (100 claims/subclaims).

lated as above) on the development set.[13]

**Results: Including context from evidence sentences improves retrieval performance.** Table 6 reports the performance of the baseline BM25, best off-the-shelf model from Table 4 (T5-3B on ANLI), and fine-tuned entailment models. Performance of models fine-tuned with evidence context on WICE is shown in the bottom half of the table. We find that these latter category of models perform best, with the best performance reported by the T5-3B model fine-tuned on WICE.[14]

### 3.3 Entailment Classification using Retrieval

Here, we use **retrieve-then-predict** rather than the MAX "stretching" strategy from Section 3.1.

**Setup** We retrieve the top-$k$ ($= 7$, in our experiments[15]) sentences using the sentence-level retrieval scores in Section 3.2.[16] These sentences are concatenated to construct a new premise/evidence; this is used by the chunk-level NLI model to make a document-level judgment: $sc(c_i, E) = P(y = \text{entailed} \mid c_i, e_{i,1} \dots e_{i,k})$, where $e_{i,1} \dots e_{i,k}$ are the top-$k$ retrieved sentences. This strategy is similar to Nie et al. (2019).

**Results** We report the performance of the chunk-level T5-3B model fine-tuned on ANLI+WICE in Table 7. It shows that the retrieve-then-predict strat-

egy using the retrieval model without evidence context does not work well. However, adding context improves performance significantly. This mirrors our evidence retrieval results from Table 6. As an upper bound, we report results in the **oracle setting**, i.e., if a gold set of supporting sentences is provided as input to the NLI model.[17] We see substantially improved performance in the oracle setting (71.1 vs 78.0 in oracle). **The large gaps suggest that better retrieval can improve the entailment classification performance**.

### 3.4 Entailment Classification by GPT

Table 8 shows the entailment classification performance of GPT-3.5 (Brown et al., 2020; Ouyang et al., 2022) and GPT-4 (OpenAI, 2023) on WICE with **oracle** retrieval, which uses the gold set of supporting sentences.[18] We use a few-shot prompt with three examples (Appendix H). We see that GPT-4 is stronger at entailment classification than our best fine-tuned subclaim-level model, but claim-level classification, which is expected to be more complex, is still challenging for simple few-shot prompting even with the oracle retrieval.

Although we evaluate the GPT models with oracle retrieval, we believe that future work can explore how to scale GPT-4 to work over long-document entailment settings (e.g., tradeoffs of using stretching vs. feeding in contexts up to the maximum size allowed by GPT-4) and reduce its cost so it can be practically deployed for fact verification in real workflows.

---

[13]When no supporting sentence is retrieved for a claim/subclaim, we define the precision, recall, and F1 for this case as zero.

[14]In the ANLI+WICE setting, we pre-trained the model on ANLI, which does not have the evidence context, and further trained it on WICE, which has the evidence context. This mismatch may be a reason for the drop in performance.

[15]98% of claims in the development set have equal to or less than 7 gold supporting sentences.

[16]For retrieval, we use T5-3B finetuned on ANLI+WICE w/o evidence context and T5-3B finetuned on WICE w/ evidence context, which are the best models in each setting.

---

[17]To avoid biases caused by the input length, we add or remove sentences in the oracle setting to make input evidences up to but no longer than 256 tokens. For the non-supported cases with no gold set of supporting sentences, we default to the MAX setting from Section 3.1. Refer to Appendix G for details of the oracle retrieval.

[18]We use first 100 claims/subclaims from the oracle retrieval dataset because the GPT-4 evaluation is costly.

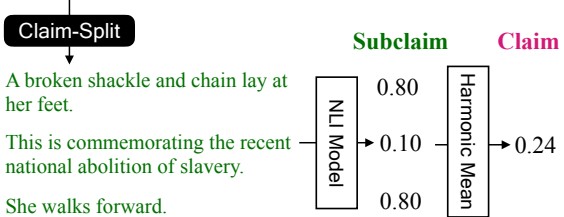

Figure 3: Entailment classification using `Claim-Split`. We decompose a claim into subclaims, predict subclaim-level entailment scores, then aggregate these to get the score for the claim.

## 4 Is `Claim-Split` useful for entailment classification?

We introduced `Claim-Split` as a method to provide fine-grained annotation by decomposing claims into simpler subclaims, and we used it with human verification to ensure the quality of our dataset. However, human evaluation in Section 2.1 shows that `Claim-Split` makes mistakes less than 10% of the time, indicating its robustness. In this section, we aim to demonstrate the potential impact of `Claim-Split` beyond the dataset construction and its applicability to diverse domains. Specifically, **we test the hypothesis that `Claim-Split` makes entailment classification easier and improves the performance of NLI models.** To show this, we evaluate a strategy of entailment prediction by aggregating the subclaim level entailment score on four datasets.

**Setup** Given a claim-evidence $(c, e)$ pair and target entailment label $y$, we compare the performance of two configurations that use identical entailment models for prediction:

1. **Standard**: As is the typical mode of inference with these models, we directly predict the entailment label for the original claim $c$. The experiments in Section 3 are also in this setting.

2. **w/ `Claim-Split`**: We first predict subclaim-level entailment scores for each subclaim $c_j \in$ `Claim-Split`$(c)$. These are combined into a claim-level entailment score using harmonic mean.[19] Figure 3 describes this configuration.

---

[19]Although more intuitive, we found that aggregating using min is quite sensitive to mistakes made by the entailment models or the `Claim-Split` method. Harmonic mean was a good balance between min and arithmetic mean.

**Fine-tuned T5-3B models on WICE**

| Test Data | Train Data | Standard | w/ Claim-Split |
|---|---|---|---|
| WICE | only WICE | 82.0 | 88.0* |
| | + ANLI | 88.2 | 89.7 |

**Off-the-shelf models trained on ANLI**

| Test Data | Model | Standard | w/ Claim-Split |
|---|---|---|---|
| WICE | T5-Large | 79.2 | 83.3* |
| | T5-3B | 80.2 | 83.2 |
| VitaminC (long) | T5-Large | 80.2 | 87.2* |
| | T5-3B | 90.6 | 92.8* |
| PAWS (long) | T5-Large | 83.7 | 86.1* |
| | T5-3B | 89.5 | 88.8 |
| FRANK (XSum) | T5-Large | 86.7 | 89.7 |
| | T5-3B | 93.2 | 93.7 |

Table 9: Comparison of AUROC scores for claim-level entailment classification task using the standard and "w/ `Claim-Split`" method. Table 15 includes results in F1 and accuracy. *: improvement from the standard method is statistically significantly with p-value $< 0.05$ according to paired bootstrap test.

**Test Data** In addition to WICE, we report results on the test sets of three datasets from Honovich et al. (2022): VitaminC (fact-verification) (Schuster et al., 2021), PAWS (paraphrase) (Zhang et al., 2019), and FRANK-XSum (summarization) (Pagnoni et al., 2021). We evaluate the 500 longest claims in VitaminC and PAWS, using length as a proxy for complexity that `Claim-Split` is designed for.[20] To decompose claims using `Claim-Split`, we use a unique prompt for each dataset that includes 2-4 dataset-specific examples (Appendix D.1).

**Models** We evaluate the performance of T5 models fine-tuned on ANLI[21] for off-the-shelf settings and on WICE for fine-tune settings. For WICE, we use the MAX setting as described in Section 3.1. Note that we use the same trained models when comparing standard and w/ `Claim-Split` settings.

**Results** We report the AUROC metric for the entailment classification task in Table 9. For most model-dataset pairs, "w/ `Claim-Split`" method outperforms the standard method of using off-the-shelf models. For the smaller T5-Large model, we observe statistically significant improvements

---

[20]This increases the mean length of VitaminC from 15.5 to 37.5, and PAWS from 21.0 to 30.1. For FRANK, we use the XSum subset to restrict our analysis to one sentence claims.

[21]Honovich et al. (2022) show that T5-11B trained on ANLI worked best for many off-the-shelf settings, but we found that we could achieve competitive performance with T5-3B.

using `Claim-Split` for three datasets. This is intuitive as reducing the complexity of the problem likely benefits models with lower capability. Our results show that despite the introduction of noise (discussed in Section 2.1), **Claim-Split is effective at simplifying the entailment classification task and improving performance**. We expect that the use of better prompts and aggregation methods would lead to further improvement.

## 5    Discussion: Outstanding Challenges

**Explainable entailment.**    Our end goal is an explainable document-level entailment system that is able to localize non-factuality within claims, as our subclaims and token-level annotation allow. This requires surfacing the right evidence, which we show remains a hard problem.

**Unsupported token detection.**    Although, we did not conduct experiments on this, we believe that localization of errors is an important problem and difficult to address with existing methods (Kamoi et al., 2023). This problem should be further studied in context of decomposition techniques like `Claim-Split`.

**Better understanding of contextualization.**    Finally, we believe that the nature of contextualization remains a major unsolved problem. While decontextualizing claims is an appealing possibility (Choi et al., 2021), we found that not all claims were easy to succinctly decontextualize. For example, *The fresco is of figures...* in Figure 2 theoretically requires specifying quite a lot of information to understand exactly what fresco is being referred to, which removes some of the benefits of subclaim splitting. Our view is that subclaims-in-context is a natural unit to explore, but further work is needed to substantiate this experimentally.

## 6    Related Work

**Short-paragraph entailment**    The majority of NLI datasets have short premises and hypotheses, i.e., single-sentence (Bowman et al., 2015; Williams et al., 2018; Liu et al., 2022) or short paragraphs (Nie et al., 2020, ANLI), and involve less multi-step reasoning. There are a few exceptions; however, ContractNLI (Koreeda and Manning, 2021) is restricted to a single domain (contracts), and DocNLI (Yin et al., 2021) uses synthetic negative data (e.g., word replacement).

**Fact-verification datasets**    A separate line of datasets designed for multi-hop reasoning comes from fact verification (Thorne et al., 2018; Schuster et al., 2021). However, in practice, claims in these datasets rarely require multiple evidence sentences (Thorne et al., 2018, FEVER) or are skewed towards statements about quantities (Schuster et al., 2021, VitaminC). In recent work, Petroni et al. (2023) looked at the attribution task in the context of Wikipedia citations, but only at the coarse level of finding a better supporting document.

**Hypothesis Decomposition**    In summarization, the Pyramid method (Nenkova and Passonneau, 2004) and its recent automated variants (Shapira et al., 2019; Zhang and Bansal, 2021; Liu et al., 2023) decompose a summary into semantic content units, but is primarily aimed at understanding what content is covered rather than the reliability of that content. More recent frameworks have looked at breaking statements down into propositions (Stanovsky et al., 2018; Ernst et al., 2021; Chen et al., 2022, 2023); our approach is similar but does not rely on supervised judgments and is not restricted to token extraction. Also, purely extractive methods are not suitable for use with off-the-shelf entailment models compared to `Claim-Split`.

Work on factuality in summarization has looked at entailment of sub-sentence units like dependency arcs (Goyal and Durrett, 2020, 2021) or using question-answer pairs to isolate specific pieces of information (Wang et al., 2020; Durmus et al., 2020; Scialom et al., 2021). However, these and related frameworks like QA-SRL (He et al., 2015) are too fine-grained for our annotation scheme.

## 7    Conclusion

We collect WICE, a new NLI dataset constructed from Wikipedia. By comparing sentences in Wikipedia against cited evidence documents, we find a rich set of real-world entailment phenomena distinct from those in prior NLI datasets. We also show that decomposing complex claims into subclaims can be a valuable pre-processing step for both annotation and entailment prediction.

## Limitations

**Scope and Diversity of the Dataset**    Although we propose WICE to evaluate and improve models for evaluating real-world entailment, this dataset only includes claims in English Wikipedia articles

and evidence in the cited websites. We observe that WICE includes diverse claims and evidence, but there are many types of real-world claims that are very different from claims in WICE in both style and content, such as political claims and social media posts.

Furthermore, almost all recent language models are pre-trained on Wikipedia articles. As a result, our dataset cannot evaluate truly zero-shot performance, given that models have been exposed to this text before. However, note that pretraining does not necessarily enable a model to know whether this fact on Wikipedia is supported by this particular document; we believe that many unsupported claims in our dataset are true, just not supported by the particular evidence documents. The fact that *all* models in our experiments are pre-trained on Wikipedia, yet they do not all perform uniformly well, supports this point. Developing datasets that are based on brand-new texts is a promising direction for future work, in order to evaluate the performance in a truly zero-shot condition.

**Baseline Models**  The experiments in this paper are mainly conducted on T5-3B, which is smaller than recent large language models (LLMs). Although we evaluate GPT-3.5 and GPT-4 on the oracle retrieval dataset in Section 3.4, we do not evaluate LLMs on the full-pipeline experiments (retrieve supporting sentences from evidence articles for entailment classification, or feed the whole evidence articles to the models). Nevertheless, our dataset can provide a realistic testbed for experiments evaluating the ability of LLMs on long documents.

**Context for Claims**  Although many existing NLI datasets target short and independent claims and evidence, claims and evidence in the WICE dataset are in-context with the surrounding text. We experimentally show that the context for evidence sentences would be useful for supporting sentence retrieval. However, our experiments do not show that providing the context of claims improves the entailment evaluation performance on the WICE dataset, in spite of our observation that some claims require anaphora resolution. We hypothesize that this observation can be attributed to the nature of our dataset, where only relevant, cited evidence documents are used. Context would be much more important if in a case like Figure 2, we were attempting to substantiate these claims based on evidence documents discussing different altars

or different frescoes. These mismatches would likely be more prevalent if we used automatic retrieval to find relevant documents. Instead, the linked documents from Wikipedia that perform the basis of our dataset are implicitly about the same entities as in the claims, so our models do not need to understand the context as thoroughly to evaluate the entailment relationships. Further work can explore broadening our findings to open-domain settings, including an open-domain version of our dataset.

## Acknowledgments

This work was principally supported by funding from Applied Research Laboratories, The University of Texas at Austin, and UT Austin's Creating Connections program. This work was also partially supported by NSF CAREER Award IIS-2145280, Good Systems,[22] a grant from Open Philanthropy, a gift from Salesforce, Inc., and a gift from Adobe.

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

## A   Human Evaluation of `Claim-Split`

In Section 2.1, we evaluated the subclaims obtained using `Claim-Split` for completeness and correctness criteria. Figure 9 shows the instructions provided to the crowd annotators for this task. It includes three examples, one correct split of claims into subclaims, one example where the subclaims fail the completeness criterion and one where they fail the correctness criterion.

Given these instructions, each HIT asks annotators to verify 4 claims (and their corresponding subclaims from `Claim-Split`). For each tuple of context, claim, and decomposed subclaims, we ask the following questions: (1) Do all decomposed sentences correctly convey the information in the original sentence? [Yes/No]. (2) If you selected No: Which decomposed texts include mistakes? [Free Text]. (3) Decomposed sentences cover ALL information in the original sentence. [Yes/No]. (4) If you selected No: What information is missing? [Free Text]. We include (2) and (4) to improve and check the annotation quality, but we do not use the answers for these questions in the analysis.

**Characterization of `Claim-Split` errors**   We manually characterized the `Claim-Split` errors in 30 dev examples of WICE. These statistics are shown in Table 10.

Some of the mistakes are relatively simple and we expect that better prompts (few-shot examples) can fix them. For example, we observe that 30% of the mistakes are caused by removing the first or intermediate clauses. In the following example, "*Before they established themselves in the upper*

| Error Category | | % |
|---|---|---|
| Completeness | Missing Intermediate Clause | 16.7 |
| | Missing Details | 13.3 |
| | Missing First Clause | 10.0 |
| | Mistake in Parsing | 10.0 |
| | Remove Parentheses | 6.7 |
| | Missing "And" | 6.7 |
| Completeness and/or Correctness | Over-splitting | 13.3 |

Table 10: Error analysis of `Claim-Split` on WICE. We annotated 30 mistakes in the development set. Each example can be assigned more than one category or left uncategorized (33.0%).

*echelon of women's tennis*" is missing in the decomposed sentences.

> Original Sentence:
> **Before they established themselves in the upper echelon of women's tennis**, Dominique Van Roost was the only player in Belgian history to be ranked in the top ten of the ATP or WTA rankings, a mark she did not achieve until 1998 after Clijsters and Henin turned professional.
>
> Decomposed Sentences:
> - Dominique Van Roost was the only player in Belgian history to be ranked in the top ten of the ATP or WTA rankings.
> - Dominique Van Roost achieved this mark in 1998.
> - Clijsters and Henin turned professional before Van Roost achieved this mark.

Another relatively simple mistake is removing parentheses (6.7%). In the following example, decomposed sentences ignore "*(Center for Predictive Engineering and Computational Sciences)*" in the original sentence.

> Original Sentence:
> In 2009 he was appointed deputy director of the PECOS center (**Center for Predictive Engineering and Computational Sciences**) at the University of Texas.
>
> Decomposed Sentences:
> - In 2009 he was appointed deputy director of the PECOS center.
> - The PECOS center is at the University of Texas.

We expect the above mistakes and "Missing And" mistakes, which ignore some words or clauses connected by "and", could be solved by changing the prompt to include examples featuring these formats.

However, we also find errors that would be difficult to solve. For example, in the following example, the fact that *Howard lives with his wife Cerys* is missing although a decomposed sentence

*Howard's wife Cerys is a doctor* mentions his wife. `Claim-Split` with our prompt sometimes make mistakes when multiple decomposed sentences should be generated for a specific span in the original sentence; for *his wife Cerys*, decomposed sentences about two facts *Howard lives with his wife Cerys* and *Howard's wife Cerys is a doctor* should be generated.

> Original Sentence:
> Howard lives in Camden, London **with his wife Cerys**, a doctor, and their dog, a Jack Russell Terrier named Archie.
>
> Decomposed Sentences:
>
> - Howard lives in Camden, London.
> - Howard's wife Cerys is a doctor.
> - Howard and Cerys have a dog named Archie.
> - Archie is a Jack Russell Terrier.

Another challenging type of mistake is caused by over-splitting. In the following example, as a result of decomposition, the information of "*after*" is lost. In this case, a candidate of correct decomposition is not to decompose the latter two sentences as "*Shine Limited was set up by former BSkyB executive Elisabeth Murdoch* **after** *she quit as broadcaster*".

> Original Sentence:
> The production company that was selected was Shine Limited, which was set up by former BSkyB executive Elisabeth Murdoch **after** she quit as broadcaster.
>
> Decomposed Sentences:
>
> - The production company that was selected was Shine Limited.
> - Shine Limited was set up by former BSkyB executive Elisabeth Murdoch.
> - Elisabeth Murdoch quit as broadcaster.

We note that we have manually fixed these mistakes in the dev and test set of the WICE dataset.

## B  Additional Data Collection Details

**Base Dataset**   As mentioned in Section 2.3, we use the same articles-claims pairs from Wikipedia as the SIDE dataset (Petroni et al., 2023) but do not use their annotations or any other aspects of their pipeline. We re-retrieve the citations from Wikipedia directly because SIDE only contains one supporting evidence even when there are multiple in the raw data. For our dataset, we use the August 2019 version for both Wikipedia and Common Crawl. We automatically parse the cited articles' HTML to extract the article text. This process is often quite noisy and may include extraneous sentences like *"Click for more"* that are not part of the

main article body; however, we included them in WICE because real-world entailment classification often requires dealing with noisy data.

Note that for claim sentences with multiple citations, we only include claims with 1 or 2 citations positioned at the end of the sentence. Cases with larger numbers of citations are infrequent (approximately $8.1\%$) and typically represented either lists or multiple articles all in support of the same base fact.

**Additional Filtering**   In Section 2.3, we outlined additional filtering using a `RoBERTa-Large` model to filter out trivially entailed claims, i.e., those for which all subclaims are predicted as entailed. Here, we provide more details of our process.

We use a pre-trained model fine-tuned on the DocNLI dataset provided by the authors of the dataset.[23] To deal with the long WICE evidences, we split the documents into chunks of less than 256 tokens each. We predict entailment scores for each chunk-subclaim pair using the NLI model. For aggregation across chunks, we classify a subclaim as entailed if it is classified as entailed by at least one chunk.

**Task Interface**   Figure 4 shows our annotation interface. The left panel shows the evidence articles which are split into sentences and numbered. The right panel shows the claim along with its preceding context. In the bottom half of the right panel, the sublclaims derived using `Claim-Split` are shown; all annotation is performed for these.

Each HIT includes annotation of one claim, i.e., 2-6 subclaims. The median work time for each HIT was about 5 minutes ($9/h).[24] For each subclaim, the annotators first select the entailment classification label and, if applicable, the supporting sentences. If they select "Partially Supported / Not Supported" in the first step, the annotation interface for non-supported tokens is shown to them (see Figure 5). For these cases, we consider a subclaim as `NOT-SUPPORTED` if the annotator highlights all subclaim tokens, else `PARTIALLY-SUPPORTED`. The "Confirm" button allows annotators to move to the next subclaim.

We build our annotation interface based on the FALTE annotation tool (Goyal et al., 2022).

---

[23] https://github.com/salesforce/DocNLI

[24] We aimed for $15/hr but workers took longer than we expected on the task. Annotators viewed our task favorably and completed it promptly, but we will strive to calibrate our pay estimates better in the future.

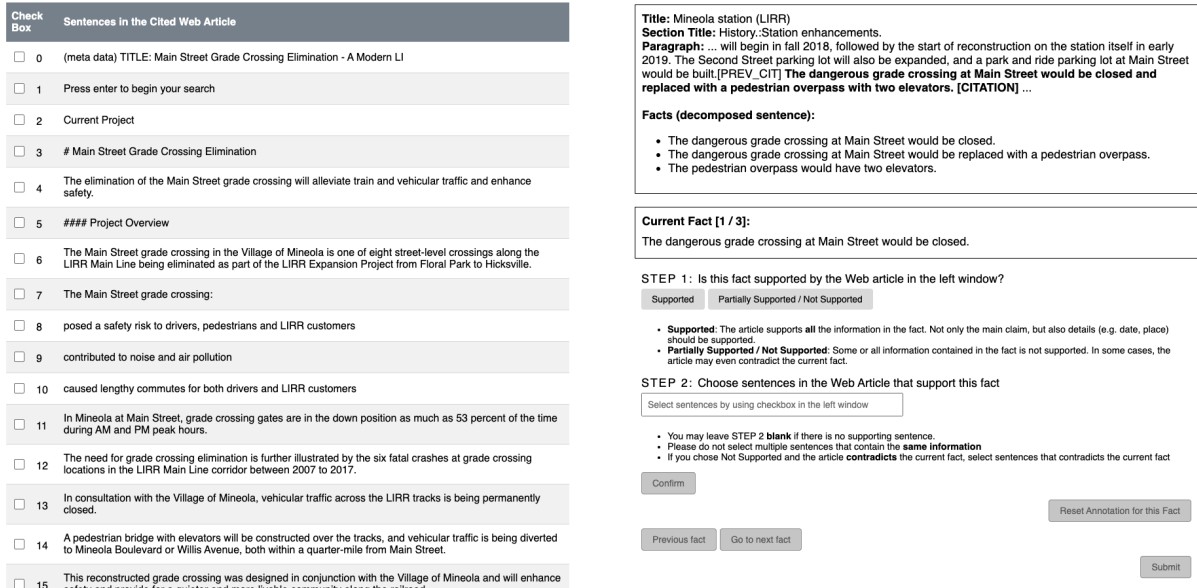

Figure 4: Annotation interface for WICE annotation.

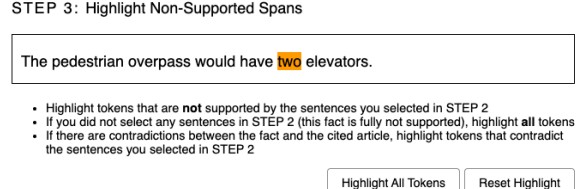

STEP 3: Highlight Non-Supported Spans

The pedestrian overpass would have **two** elevators.

- Highlight tokens that are **not** supported by the sentences you selected in STEP 2
- If you did not select any sentences in STEP 2 (this fact is fully not supported), highlight **all** tokens
- If there are contradictions between the fact and the cited article, highlight tokens that contradict the sentences you selected in STEP 2

Highlight All Tokens    Reset Highlight

Figure 5: Annotation interface for highlighting non-supported tokens.

**Filtering Steps** We perform filtering at different stages of our dataset collection process, e.g., while retrieval of evidence articles from Common Crawl, cleaning the base dataset or during filtering using the NLI model. Table 11 shows the number of data points removed at each step of the filtering process for the development set.

**Aggregation of subclaim level labels from different workers** For **entailment classification**, we take a majority vote between worker labels. If no majority exists but 2 annotators each select PARTIALLY-SUPPORTED and NOT-SUPPORTED for the dev or test set, we choose PARTIALLY-SUPPORTED as the final label. In all other scenarios, we remove the subclaim (and the corresponding claim $c$) from our dataset as these cases tend to be quite subjective. This filters out 12.5% of the claims in the development set.

For **supporting evidence sentences**, we retain individual sets of supporting sentences by all workers who chose the majority entailment label (the

| Filtering Step | %removed | #post-filtering |
|---|---|---|
| Before Filtering | | 4,545 |
| Missing Wikipedia pages | 7.1 | 4,222 |
| Bullet points | 7.5 | 3,905 |
| Cite not at sentence end | 17.1 | 3,238 |
| Prev cite not at sentence end | 13.4 | 2,805 |
| #Citations >= 3 | 8.1 | 2,577 |
| Missing in Common Crawl | 24.3 | 1,951 |
| HTML postprocessing failed | 14.9 | 1,661 |
| #subclaims = 1 or > 6 | 19.1 | 1,343 |
| RoBERTa-large classified all subclaims as entailed | 45.0 | 739 |

Table 11: Statistics of the filtering of the development set. The original size is 4,545 and the final size is 739. Note that we did not annotate all these data and the number of claims in development set is 349.

label selected in the above step). We prefer this over union as there can be multiple different sets of sentences with identical information (e.g., the date can sometimes be ascertained from several different sentences). We found that this subset of workers chose the exact same set of supported sentences for 56.1% of SUPPORTED cases and 34.4% of PARTIALLY-SUPPORTED cases, which shows high inter-annotator agreement.

To aggregate **unsupported tokens** in subclaims, we take a token-level union of all workers who chose PARTIALLY-SUPPORTED as the entailment label. For this task, we remove data points if any annotator disagrees with the final set of tokens by more than three tokens (25.3% PARTIALLY-SUPPORTED subclaims in the develop-

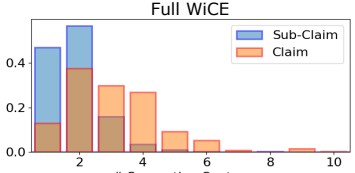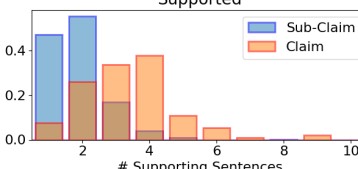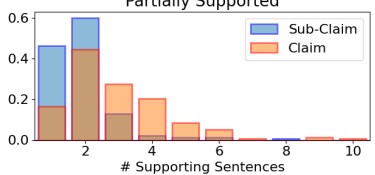

Figure 6: Distribution of the number of supporting sentences for each claim and subclaim in the development set of WICE. The figures shows this distribution for PARTIALLY-SUPPORTED, SUPPORTED and the combined set.

|  | Train | Dev | Test |
|---|---|---|---|
| # Partially Supp. subclaims | 374 | 124 | 105 |
| # Tokens / subclaim | 12.8 | 13.0 | 14.4 |

Table 12: Statistics of WICE for non-supported token annotation. These annotations are for partially supported subclaims with high inter-annotator agreement. Each subclaim in the development set includes 3.3 non-supported tokens (25.2% of tokens) on average.

ment set).

## C Additional Dataset Statistics and Analysis

**Supporting Sentences**  Figure 6 shows the distribution of the number of supporting sentences f PARTIALLY-SUPPORTED, SUPPORTED and the combined set in WICE. Supported and partially supported subclaims have almost the same number of supporting sentences annotated (1.9 on average), but supported claims have more supporting sentences annotated compared to partially supported claims (averages are 3.4 and 2.9).

**Non-Supported Tokens**  After filtering out data points with low inter-annotator agreement on the annotation for non-supported tokens, there are 374 partially supported subclaims in the training data with an average of 12.8 tokens per subclaim in (Table 12). Partially supported subclaims have 3.3 non-supported tokens (25.2% of tokens in subclaims) on average in the development set.

**Word Overlap**  Figure 7 shows the word overlap between claims and evidence (the recall of claim bigrams that are also in the evidence) for WICE, VitaminC, and FEVER. Although VitaminC manually created claims so that this overlap is lower, we observe that the real claims in WICE have competitively low claim-evidence overlap. Furthermore, examples corresponding to different entailment labels have similar word overlaps, especially at the

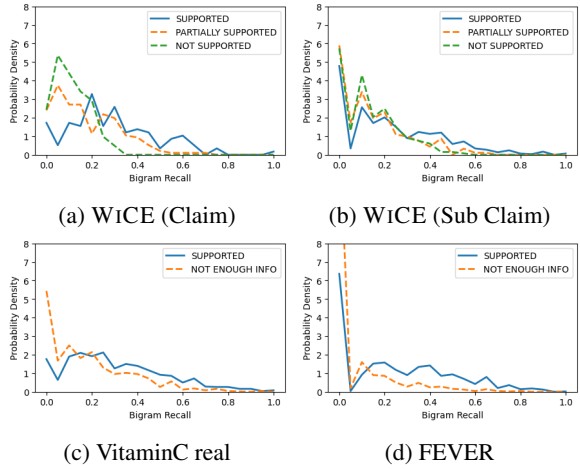

(a) WICE (Claim)  (b) WICE (Sub Claim)

(c) VitaminC real  (d) FEVER

Figure 7: Word overlap (recall of bigrams) between claims and evidence.

subclaim level. This suggests that WICE does not suffer from spurious biases.

**Analysis of Phenomenon**  In Section 2.5, we categorized entailment relationships between claims and evidence into several categories. We provide examples for each category in Table 13. Note that the FEVER example, a *character* is a *person*, represents a typical type of the inferences required in this dataset, which are simple hypernymy.

Additionally, we also characterize distribution of semantic roles that form each claim. We use a taxonomy drawn from Davidsonian event semantics (Truswell, 2019) and semantic roles to characterize what each claim is about: this consists of *events*, *properties* (attributes describing participants in an event; these include *name*, *occupation*, *nationality*, *quantity*, *ordinal*), *location*, *time*, *reason* (why something happened), *manner*, and *evidentials*. Each claim may carry multiple labels, and subclaims consist of not necessarily disjoint subsets of these categories, e.g., "Jones buttered his toast slowly" (*event*, *manner*) and "Jones buttered his toast in the bathroom" (*event*, *location*). Table 14 shows this distribution.

| Verification Category | Dataset | Claim | Evidence |
|---|---|---|---|
| Compression | WICE | Brian Wilkins had been using the beach since 1938. | ... And cutting the ribbon will be **Brian Wilkins**, 82, **who has been using the beach since 1938** and been an enthusiastic supporter of the campaign to get it back open. ... |
| Compression w/ contextualization | WICE | **His actions** were on December 26, 1944, in the vicinity of Sommocolonia, Italy. | ... First Lieutenant John R. Fox distinguished himself by extraordinary heroism at the risk of his own life on **26 December 1944** in the Serchio River Valley Sector, **in the vicinity of Sommocolonia, Italy**. ... |
| Paraphrase - Direct | WICE | The Chicago Board of Trade is the largest and most diverse derivatives market **globally**. | Chicago has one of the **world's** largest and most diversified economies, with more than four million employees and generating an annual gross regional product (GRP) of over $609 billion. |
| Paraphrase - Require Calculation | VitC | Cases of COVID-19 have been confirmed in **more than 185** countries. | As of 22 March, more than 336,000 cases of COVID-19 have been reported in over **189** countries and territories, resulting in more than 14,400 deaths and 96,000 recoveries. |
| Paraphrase - Require Inference | WICE | Young faced Kedzie **again** in a five-round title rematch. | ... Two of MMA's top 135-pound female fighters will collide in a five-round title fight at Jackson's MMA Series 4 on April 9th in Albuquerque, New Mexico. ... **Both fighters recently took part in the ill-fated Ultimate Women Challenge reality show competition last year, though results from the fights that took place during filming have yet to be released**. ... |
| | FEVER | Wilhelmina Slater is a person. | Wilhelmina Vivian Slater ( born Wanda Slater ) is a fictional character in the American dramedy series Ugly Betty. |
| Paraphrase - Require Background Knowledge | WICE | United defeated Arsenal 5–4. | Vic Groves went close to an equaliser but the **'Busby Babes'** held out for a famous 5-4 victory. |

Table 13: Examples for categories of entailment classification problems in Table 3.

| What is being asserted? | | % |
|---|---|---|
| Event | | 52.9 |
| Time | | 3.6 |
| Location | | 10.7 |
| Manner | | 6.4 |
| Reason | | 7.9 |
| Property | Name | 6.4 |
| | Occupation | 5.7 |
| | Quantity | 12.9 |
| | Ordinal | 3.6 |
| | Other | 10.0 |
| Others | | 2.9 |

Table 14: Estimated distribution of subclaims types in WICE. Each subclaim may have multiple properties.

# D Claim-Split

This section provides further details and results of Claim-Split.

## D.1 Claim-Split Prompts

We used the following prompt template with six examples:

```
Segment the following sentence into
individual facts:
```

```
Sentence: <example claim>
Facts:
- <example subclaim>
- <example subclaim>
- ...
Sentence: <input claim>
Facts:
```

For the experiments in Section 4 on VitaminC, PAWS, and FRANK, we use the following instruction with three or four examples to generate subclaims that are suitable for the off-the-shelf evaluation by entailment classification models:

```
Please decompose the following sentence
into decontextualized sentences while
ensuring all information is retained and
the wording is as unchanged as possible
(please return the original sentence if
it cannot be decomposed):
...
```

Full prompts with few-shot examples are provided in our GitHub repository.

## D.2 Claim-Split Aggregation Performance

Table 15 shows additional results for Claim-Split aggregation experiments in Section 4. We observe

| Dataset | Model | Train Data | F1 | | Accuracy | | AUROC | |
|---|---|---|---|---|---|---|---|---|
| | | | Original | Claim-Split | Original | Claim-Split | Original | Claim-Split |
| WICE | T5-large | ANLI | 61.8 | 64.5 | 70.7 | 70.1 | 79.2 | 83.3* |
| | T5-3B | ANLI | 64.3 | 64.1 | 75.1 | 71.5 | 80.2 | 83.2 |
| | T5-3B | DocNLI | 56.4 | 62.4* | 62.8 | 70.7* | 72.5 | 77.5* |
| | T5-3B | WiCE | 65.3 | 72.7* | 77.1 | 82.4* | 82.0 | 88.0* |
| | T5-3B | ANLI+WiCE | 72.1 | 71.4 | 79.1 | 80.7 | 88.2 | 89.7 |
| VitC (long) | T5-large | ANLI | 79.2 | 81.0 | 78.0 | 81.4* | 86.7 | 90.4* |
| | T5-3B | ANLI | 86.0 | 85.2 | 85.8 | 85.6 | 91.6 | 92.6 |
| PAWS (long) | T5-large | ANLI | 70.6 | 73.2 | 72.8 | 79.6* | 81.7 | 87.2* |
| | T5-3B | ANLI | 74.1 | 75.2 | 77.2 | 79.0 | 86.2 | 89.4* |
| FRANK (XSum) | T5-large | ANLI | 40.0 | 32.2 | 95.7 | 86.1 | 86.6 | 86.0 |
| | T5-3B | ANLI | 38.0 | 46.6* | 87.9 | 92.1* | 91.1 | 91.4 |

Table 15: Claim-level entailment classification performance by aggregating scores for the subclaims generated by `Claim-Split`. `Claim-Split` improves entailment classification performance, especially on smaller T5-large models. *: p-value $< 0.05$ in paired bootstrap test.

improvement in almost all models and metrics. This result suggests that `Claim-Split` effectively reduces the complexity of the claims.

## E Implementation Details

We use PyTorch (Paszke et al., 2019) and Hugging Face Transformers (Wolf et al., 2020) libraries in our implementation. We use a machine with NVIDIA Quadro RTX 8000.

### E.1 Baseline Models

When available, we use NLI models provided by prior works and available on the HuggingFace Hub[25] for experiments in Section 3.1. The model names are provided in Table 16.

We fine-tune T5 models on VitaminC + MNLI using the dataset provided by Schuster et al. (2021). For fine-tuning on ANLI and DocNLI, we use the datasets provided by the authors.[26]

For evaluating the BM25 performance in Table 6, we use the rank_bm25 library.[27]

### E.2 Training Data of WICE

The WICE dataset provides entailment labels $y$ for each evidence-claim/subclaim pair, i.e., $(E, c, y)$. However, due to the size of the document, we need to partition it into multiple partitions $\{p_1, p_2, ...p_m\}$ (either individual sentences or concatenation of sentences of at most 256 tokens) and train on these partition-claim/subclaim pairs. To do this, we need entailment labels for each of these

---

[25]https://huggingface.co/models
[26]ANLI: https://github.com/facebookresearch/anli, DocNLI: https://github.com/salesforce/DocNLI
[27]https://github.com/dorianbrown/rank_bm25

partitions, i.e., $(p_i, c, y_{p_i})$. Here, we describe how we derive these gold labels $y_{p_i}$ using the supporting sentences annotation included in WICE.

For $(p_i, c)$, if the partition $p_i$ does not include any supporting sentence of $c$, we label $y_{p_i} =$ NOT-SUPPORTED. If the partition includes **all** supporting sentences for $c$ and the original entailment label in WICE is SUPPORTED, we label $y_{p_i}$ as SUPPORTED. For all other cases, we set $y_{p_i}$ as PARTIALLY-SUPPORTED.

### E.3 Fine-tuning T5 Models

**Input Format** The input format to our T5 models is "entailment: claim [SEP] evidence". The T5 models are trained to generate the following single-character tokens corresponding to entailment labels: e (entailed or supported), p (partially supported), n (not supported or neutral), and c (contradiction). For example, fine-tuning on WICE uses e, p, and n. As the model places a probability distribution over the whole vocabulary, we normalize over our target set of labels to get classification probabilities.

**Hyperparameters** We fine-tuned T5 (T5-large and T5-3B) for $25K$ steps with a learning rate of $10^{-4}$ and batch size of 32. We compute the accuracy on the development set after every $1,000$ steps and save the best model checkpoint. We use a maximum input length of $512$ tokens during training, but feed all input tokens during inference.

Training with input size $512$ is not suitable for DocNLI, which includes many premises longer than $512$ tokens. We use the input size of $512$ following Yin et al. (2021). Although this is the best

| Model | Train Data | Model Name |
|-------|-----------|------------|
| RoBERTa-large | SNLI | boychaboy/SNLI_roberta-large |
| RoBERTa-large | MNLI | roberta-large-mnli |
| ALBERT-xlarge | VitaminC | tals/albert-xlarge-vitaminc |
| ALBERT-xlarge | VitaminC+MNLI | tals/albert-xlarge-vitaminc-mnli |

Table 16: Publicly available models in Hugging Face Hub used in our experiments in Section 3. VitaminC models are provided by Schuster et al. (2021). These models are also used in experiments in Laban et al. (2022).

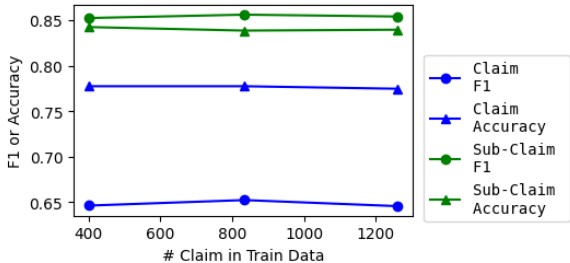

Figure 8: Relation between the size of train set and performance on the entailment classification task of WICE on T5-3B.

setting in their experiments, this is a possible reason for the lower performance of the model finetuned on DocNLI compared to the ANLI model.

## F Effect of Training Data Size on Entailment Classification Performance

Figure 8 shows the performance of T5-3B finetuned on different amounts of training data from WICE. This result suggests that the performance of fine-tuning on WICE is saturating with this training procedure and models we used in this paper, or that **much** larger amounts of training data are needed to improve the performance.

## G Oracle Retrieval Dataset

WICE is a dataset that is designed for evaluating entailment classification on long evidence articles, so it requires supporting sentence retrieval as a first step for language models that cannot evaluate very long inputs. To solely evaluate the entailment classification capability of language models, we make the oracle retrieval dataset, which is designed to simulate the situation in which we have an ideal retrieval model. This dataset provides a gold set of supporting sentences as input to NLI models. The oracle retrieval dataset is used in experiments in Section 3.3 and 3.4.

The oracle retrieval dataset consists of oracle chunks that include all sentences in a gold set of supporting sentences.[28] We note that there can be multiple oracle chunks for each claim/subclaim because different annotators can annotate different sets of supporting sentences, which include sufficient information to support (or partially support) the claim/subclaim.

To avoid biases caused by the number of supporting sentences (e.g. supported claims may have a larger number of supporting sentences than partially-supported claims), we add randomly selected sentences from the evidence article to the oracle chunks. Specifically, we add randomly selected sentences until the size of the chunks reaches 256 tokens (the chunks are equal to or shorter than 256 tokens).

For non-supported cases, which do not have any gold supporting sentences, we provide chunks as in the MAX setting in Section 3.1. The chunks in this setting also include at most 256 tokens in most cases.[29]

Finally, to avoid biases caused by the number of oracle chunks, we randomly select three oracle chunks for each claim/subclaim. A straightforward way of performing entailment classification on the oracle retrieval dataset is to evaluate the entailment score for every oracle chunk for each claim/subclaim and take the maximum entailment score. Therefore, claims/subclaims with a large number of oracle chunks are likely to receive higher entailment scores. To avoid this bias, we make every claim/subclaim in the oracle retrieval dataset has the same number of oracle chunks.

## H Entailment Classification by GPT

We provide details of the entailment classification experiment on GPT-3.5 and GPT-4 in Section 3.4.

---

[28]When the oracle chunk becomes longer than 256 tokens only with the gold supporting sentences, we remove gold supporting sentences to make the chunk shorter than or equal to 256 tokens.

[29]We do not split sentences into sub-sentences when we make chunks. Therefore, when an evidence article includes a sentence longer than 256 tokens, we include the sentence in the chunk as is.

**Models** We use `gpt-3.5-turbo-0613` and `gpt-4-0613` for this experiment.[30]

**Dataset** We use the first 100 claims/subclaims of the oracle retrieval dataset (Appendix G) for this experiment.

**Prompts** We provide the prompt used in this experiment below. We use XML as an output format as in (Das et al., 2023) to make post-processing easy. Our prompt includes three examples from the development set of WICE, but we omitted two examples and evidence sentences in the first example. Our GitHub repository includes the full prompt.

> Your task is to evaluate if a claim is supported by a provided evidence article snippet.
>
> You need to present your explanation first, and then choose your conclusion from the options [supported, partially_supported, not_supported].
>
> We provide several examples. Your response must be in the same format as the XML in the examples.
>
> Examples:
> <input>
> <claim>On August 22, 2017, Richard Amardi was selected to play for the Canadian Senior Men's National Team to compete in the FIBA AmeriCup 2017 in Argentina.</claim>
> <evidence>
> <sentence_39>Values, Vision and Missions</sentence_39>
> <sentence_40># SENIOR MEN'S NATIONAL TEAM ANNOUNCES FIBA AMERICUP 2017 ROSTER</sentence_40>
>
> (Evidence Sentences are Omitted)
>
> </evidence>
> </input>
> <!– Your explanation and answer should be written below –>
> <output>
> <explanation>Sentence 41 says that the final roster for the Senior Men's National Team set to compete at the FIBA AmeriCup 2017 in Argentina was announced on August 22, 2017. Sentence 77 shows that Richard Amardi is on the list.</explanation>
> <answer>supported</answer>
> </output>
>
> (Two examples are omitted)
>
> Here is your task:
>
> <input>
> <claim>{claim}</claim>
> <evidence>
> {evidence}
> </evidence>
> </input>
> <!– Your explanation and answer should be written below –>

---

[30] https://platform.openai.com/docs/models

## I  Datasheet for WICE

In this section, we provide a datasheet (Gebru et al., 2021) for the WICE dataset.

### I.1  Motivation for Datasheet Creation

The information regarding the individuals or organizations who created or funded the dataset will be included in the camera-ready version.

**For what purpose was the dataset created?** There are some major challenges when applying modern entailment models to measure real-world attribution and factuality consistency. Specifically, existing natural language inference (NLI) models and datasets target relatively short claims and evidence, negative examples are often artificially created, and fine-grained labels have not been studied well. We create the WICE dataset to address these limitations in the existing NLI datasets.

### I.2  Dataset Composition

**What are the instances?** Each instance in WICE is a group of subclaims derived from a claim (a sentence in a Wikipedia article) and evidence (cited websites).

**Is there a label or target associated with each instance?** The annotation for each subclaim includes the entailment label (supported, partially-supported, or not-supported), supporting sentences (a subset of evidence sentences that support or partially support the subclaim), and non-supported tokens (tokens in the subclaim that are not supported by the evidence).

**How many instances are there?** WICE includes 1,260, 349, and 358 claims in the train, development, and test data, which are decomposed into 3,470, 949, and 958 subclaims. Detailed dataset statistics are provided in Table 1.

**Does the dataset contain all possible instances or is it a sample of instances from a larger set?** The claims in WICE are a sub-set of sentences in Wikipedia articles. The sentences are randomly selected from those used in the SIDE dataset (Petroni et al., 2023).

**Is the dataset self-contained?** Yes, all resources are included in our release.

### I.3  Data Collection Process

**How was the data associated with each instance acquired?** We acquired claims from Wikipedia

and evidence from Common Crawl (the August 2019 version).

**Who was involved in the data collection process and how were they compensated?** The labels (the entailment classification, supporting sentences, and non-supported tokens) are annotated by workers recruited using Mechanical Turk. Each instance is annotated by five workers. Annotators were paid $0.75 per claim annotation.

**Over what timeframe was the data collected?** Claims and evidence are collected from the August 2019 version of Wikipedia and Common Crawl. The annotation by workers was conducted in November and December 2022.

### I.4 Data Preprocessing

**What preprocessing / cleaning was done?** We automatically parse the cited articles' HTML to extract the article text. In addition, we decompose claims (sentences in Wikipedia) by using `Claim-Split` (Section 2.3). Details of the filtering process are described in Appendix B.

**What software was used to preprocess the data?** We use Beautiful Soup 4 to extract sentences from the HTML of the retrieved articles. We use GPT-3.5 (Brown et al., 2020; Ouyang et al., 2022) to decompose claims into subclaims. Specifically, we use the `text-davinci-002` model through the OpenAI API in November 2022.

### I.5 Dataset Distribution

**How will the dataset be distributed?** The WICE dataset is available in a GitHub repository.[31]

**Who will be supporting and maintaining the dataset?** This dataset will be maintained by the authors of this paper.

---

[31] https://github.com/ryokamoi/wice

**Instructions**

In this task, you will be shown a long, complex sentence. Additionally, you will also be shown a list of shorter sentences that should convey the same information as the complex sentence. Your task is to judge whether the list of shorter claims (a) CORRECTLY convey the information from the complex sentence, and (b) COVER ALL the information from the complex sentence.

**Good Example**

*Here, the 1st sentence can be further decomposed, but you do not have to point out this problem.*

*Context:*

On the next day, 1 more coffee shop was opened. The first Starbucks store in Slovakia opened in Aupark Shopping Center in Bratislava on May 31, 2016, with two more stores confirmed to open in Bratislava by the end of 2016. In February 2016, Howard Schultz announced the opening of stores in Italy. The first Italian Starbucks store was inaugurated in Milan on September 6, 2018. After Taste Holdings acquired outlet licensing for South African stores, Starbucks opened its first store in South Africa in Rosebank, Johannesburg on Thursday, April 21, 2016, and its second in the country at the end of April in Mall of Africa. In May 2017, Starbucks announced it was commencing operations in Jamaica, where the first store is to open in the resort city of Montego Bay.

*Original Sentence:*

The company announced that its first store would be on located on the shores of the world-famous Doctor's Cave Beach, offering views of the Caribbean Sea.

*Decomposed Sentences:*

1. The company announced that its first store would be located on the shores of the world-famous Doctor's Cave Beach.
2. The store would offer views of the Caribbean Sea.

**Bad Example 1.**

*Here, the 1st sentence is incorrect because she was NOT the only member of the Washington Health Benefit Exchange 2, although she was the only one who voted against the salary increase.*

*Context:*

Teresa Mosqueda is an American politician and labor activist from Seattle, Washington. She was elected to the Seattle City Council in 2017 to represent the at-large position 8.

*Original Sentence:*

In November 2013, she was the only member of the Washington Health Benefit Exchange who voted against increasing the salary of the health exchange's CEO by 13%.

*Decomposed Sentences:*

1. In November 2013, she was the only member of the Washington Health Benefit Exchange.
2. She voted against increasing the salary of the health exchange's CEO by 13%.

**Bad Example 2.**

*Here, all individual claim are correct but they do not cover all the information from the complex sentence. Here, it does not include the fact that Howard lives with his wife.*

*Context:*

(no context)

*Original Sentence:*

Howard lives in Camden, London **with his wife Cerys**, a doctor, and their dog, a Jack Russell Terrier named Archie.

*Decomposed Sentences:*

1. Howard lives in Camden, London.
2. Howard's wife Cerys is a doctor.
3. Howard and Cerys have a dog named Archie.
4. Archie is a Jack Russell Terrier.

Figure 9: Instructions given to crowd annotators to evaluate the correctness and completeness of subclaims generated using the `Claim-Split` method.