# OpenReview forum: "WiCE: Real-World Entailment for Claims in Wikipedia"
_EMNLP/2023/Conference — EMNLP 2023 Main_

### Official Review · Reviewer_rBnT · 2023-07-30

**Soundness:** 4

**Excitement:**

4: Strong: This paper deepens the understanding of some phenomenon or lowers the barriers to an existing research direction.

**Paper Topic And Main Contributions:**

This paper introduces a TE dataset built on natural claim and evidence pairs extracted from Wikipedia; especially annotations over sub-sentence units. The authors conduct sufficient NLP experiments to support claims about shortcomings of prior NLI datasets and the usefulness of the dataset.

**Questions For The Authors:**

- what is the average hourly for m-turkers if you pay 0.75$ per HIT?

**Reasons To Accept:**

- good presentation and well written
- paper identifies use-cases of models trained on Textual entailment and finds that the underlying datasets are not suited for such use cases. Hence, they propose a new dataset for this.
- I think such a dataset is a useful and interesting contribution to the community
- good amount of relevant experiments conducted


**Reasons To Reject:**

- None really

**Reproducibility:**

4: Could mostly reproduce the results, but there may be some variation because of sample variance or minor variations in their interpretation of the protocol or method.

**Reviewer Confidence:**

3: Pretty sure, but there's a chance I missed something. Although I have a good feel for this area in general, I did not carefully check the paper's details, e.g., the math, experimental design, or novelty.

---

> ### Author Rebuttal · Authors · 2023-08-27
>
> We sincerely appreciate your time in reading and reviewing our paper.
>
> > what is the average hourly for m-turkers if you pay 0.75$ per HIT?
>
> We aimed for \\$15/hr but workers took longer than we expected on the task. The median work time per HIT was about 5 minutes, so the median payment was approximately \\$9/hr (line 1207). We calculated the median because some workers took a very long time and the average is affected by these outliers. Annotators viewed our task favorably and completed it promptly, but we will strive to calibrate our pay estimates better in the future.

---

### Official Review · Reviewer_rRJ9 · 2023-08-03

**Soundness:** 4

**Excitement:**

4: Strong: This paper deepens the understanding of some phenomenon or lowers the barriers to an existing research direction.

**Missing References:**


[1] Logic-Regularized Reasoning for Interpretable Fact Verification. AAAI 2022.

**Paper Topic And Main Contributions:**

This paper proposes WICE, a new fine-grained textual entailment dataset built on natural claim and evidence pairs extracted from Wikipedia.
It provides entailment judgments over subsentence units of the claim, and a minimal subset of evidence sentences that support each subclaim.
Experiments show that decomposing complex claims into subclaims can be a valuable pre-processing step for both annotation and entailment prediction.



**Reasons To Accept:**


1. Fine-grained textual entailment is more realistic and deserves further research.
2. Paper is well-written and the experiments are solid.



**Reasons To Reject:**


1. The data comes from Wikipedia, which makes it difficult to test the model's capabilities, as many models are pre-trained on the Wikipedia.



**Reproducibility:**

4: Could mostly reproduce the results, but there may be some variation because of sample variance or minor variations in their interpretation of the protocol or method.

**Reviewer Confidence:**

3: Pretty sure, but there's a chance I missed something. Although I have a good feel for this area in general, I did not carefully check the paper's details, e.g., the math, experimental design, or novelty.

---

> ### Author Rebuttal · Authors · 2023-08-27
>
> We sincerely appreciate your time in reading and reviewing our paper.
>
> > The data comes from Wikipedia, which makes it difficult to test the model's capabilities, as many models are pre-trained on the Wikipedia.
>
> As you correctly pointed out, most language models have been exposed to the Wikipedia articles used in WiCE during pretraining. However, the models have not necessarily been exposed to the evidence articles, and even if they were, pretraining does not necessarily enable a model to know whether this fact on Wikipedia is supported by this particular document. Therefore, we believe that pre-training on Wikipedia articles does not strongly influence their performance on our dataset, except insofar as Wikipedia data is comfortably in domain for pre-trained LMs. In fact, our experiments showed that our dataset is challenging for the T5 models.
>
> We agree that developing datasets that are based on brand-new texts is a promising direction for future work, in order to evaluate the performance in a truly zero-shot condition.
>
> > [1] Logic-Regularized Reasoning for Interpretable Fact Verication. AAAI 2022.
>
> Thank you for pointing out the missing reference. We agree that this work in interpretable fact verification is related to our work, particularly with regard to Claim-Split and non-supported tokens detection. We will include this reference in the final version of our paper.

---

### Official Review · Reviewer_SxkM · 2023-08-03

**Soundness:** 3

**Excitement:**

4: Strong: This paper deepens the understanding of some phenomenon or lowers the barriers to an existing research direction.

**Paper Topic And Main Contributions:**

This paper is about a new entailment dataset created from Wikipedia.
Their dataset has three key features compared to existing similar datasets;
(1) its premises are long;
(2) negative examples are natural ones, i.e. not synthetic ones; and
(3) its annotation is fine-grained and allows to identify which parts of claims are not supported by evidence text.

Quantitative analyses in Table 3 shows that their dataset is more challenging than previous datasets (FEVER and VitaminC).

The dataset allows to conduct three entailment-related tasks;
(a) entailment classification;
(b) evidence retrieval; and
(c) non-supported token detection.

The authors demonstrated that these three tasks defined on their dataset were difficult by comparing
the performances of several competitive models with those of humans.

The authors also proposed Claim-Split, which automatically splits a claim into sub-claims using GPT-3.5.
Their experiments showed that Claim-Split was effective for entailment classification.


**Reasons To Accept:**

The new entailment dataset will be useful for textual entailment recognition research.

A series of the quantitative analyses and experiments are informative for RTE researchers.


**Reasons To Reject:**

The weakness of this paper is the lack of description and justification for some important aspects of the paper.

First, the contributions of the paper are not clearly described.
I guess they are (1) the new entailment dataset that has the above three features, (2) the Claim-Split method, and
(3) the quantitative analyses and experiments.

Second, it should be justified why the authors did not directly use GPT-3.5 for the three tasks (entailment
classification, evidence retrieval, and non-supported token detection). The authors used GPT-3.5 for Claim-Split.
Their problem settings therefore allow to use GPT-3.5. It should be a reasonable baseline given that we can use it
relatively easily.

Third, 50 samples might be too few to draw any conclusion about human performances for the tasks.
Using only 50 samples should have been justified too.

Fourth, I wonder why AUROC was used for the entailment classification results in Table 8 while
F1 and accuracy were used for the entailment classification results in Tables 4 and 5. This needs explanation.

Last, there are many descriptions that need more detailed explanations or justification.
I understand that there may be many details that cannot be written in the paper.
But some of them seem important information or key research decisions. Below are some examples:

p.1

* What does it mean for a token to be supported? A fact or a proposition can be supported.
But I don't really understand how a token is supported. Do you simply mean that a token is supported
if the token is written in an evidence text?

* What do you mean by "ecologically valid"? Do you mean naturally occurring negative examples?

p.3

* "In total, only 8.6% of the claims included one of the two types of errors."
... What are the two types?

* "we re-retrieve the cited web article(s)"
... Why did you need to re-retrieve them?

* "Also, we filter claims that are decomposed into either only one or more than six subclaims."
... Why did you need this filter? Why six?

p.4

* In Table 3, WiCE Subcl and WiCE Claim do not add up to 100%.

p.6

* "p(SUPPORTED) + 0.5 × p(PARTIALLY-SUPPORTED)"
... How was this derived?

* "As there can be multiple gold sets of supporting sentences for each claim/subclaim in WICE,"
... I could not understand this clause.

* In Table 6, why was ANLI+WiCE worse than WiCE in the "w/ evidence context" setting, while
it was better in the other setting?

p.7

* I could not understand footnote 7.


**Reproducibility:**

3: Could reproduce the results with some difficulty. The settings of parameters are underspecified or subjectively determined; the training/evaluation data are not widely available.

**Reviewer Confidence:**

4: Quite sure. I tried to check the important points carefully. It's unlikely, though conceivable, that I missed something that should affect my ratings.

**Typos Grammar Style And Presentation Improvements:**

p.2

"Real world claims, such are those"
-->
"Real world claims, such as those"

p.6

"Performance of models fine-tuned with evidence context on WICE is supported in the bottom half of the table."
-->
"Performance of models fine-tuned with evidence context on WICE is reported in the bottom half of the table."

---

> ### Author Rebuttal · Authors · 2023-08-27
>
> We appreciate your detailed comments and valuable suggestions!
>
> > First, the contributions of the paper are not clearly described. I guess they are (1) the new entailment dataset that has the above three features, (2) the Claim-Split method, and (3) the quantitative analyses and experiments.
>
> Yes, that is a precise summary of our contribution. We will make it clearer in the final version of the paper.
>
> > Second, it should be justified why the authors did not directly use GPT-3.5 for the three tasks
>
> Thank you for your suggestion. In this work, we focused on evaluating the models fine-tuned on NLI datasets because prior work (e.g., Gao et al., 2021; Wei et al., 2022; Yue et al., 2023) showed that LLMs often do not work well on related entailment tasks in a zero-shot or few-shot condition, as we mentioned in the limitations section (line 610). In addition, using LLMs for very long context inputs would be costly.
>
> However, since we finalized the version of this work submitted to EMNLP, OpenAI has continued to reduce prices for its models, and also increased the maximum input lengths. We will conduct benchmarking of LLM methods in any future version.
>
> > Third, 50 samples might be too few to draw any conclusion about human performances for the tasks. Using only 50 samples should have been justified too.
>
> Note that in Tables 4 and 5, the gap between human performance and model performance is quite large. With 50 samples and p = 0.92, the standard deviation of a binomial distribution is roughly 2, or roughly 4%. Even allowing for this error, the human performance is still significantly higher than model performance.
>
> These examples are time-consuming to annotate (as they require reading entire cited documents), and we wanted to conduct the annotation carefully ourselves to establish a realistic human upper bound.
>
> We also note that we annotated 144 sub-claims because we annotated 50 claims in WiCE, so the number of subclaims is substantially higher.
>
> > Fourth, I wonder why AUROC was used for the entailment classification results in Table 8 while F1 and accuracy were used for the entailment classification results in Tables 4 and 5. This needs explanation.
>
> For the experiment in Table 8, we used the TRUE benchmark (Honovich et al., 2022; mentioned in line 479).  Honovich et al. (2022) used AUROC in their main tables to accommodate datasets that don’t have separate development sets for tuning a classification threshold, so we used this metric here to make the results more directly comparable.
>
> However, since the datasets we used in Table 8 have both the development and open test sets, we can also calculate F1 and accuracy. We agree that it is better to also show F1 and accuracy and will add them in the final version of our paper.
>
> > What does it mean for a token to be supported?
>
> Annotators selected tokens that correspond to a sub-part of a claim that is semantically not supported by the evidence. These frequently correspond to modifiers (“in 1998”, if this is not stated) or propositions as suggested here. Our annotators had a consistent view of what this means: we observe high inter-annotator agreement, as mentioned in line 1247, and we ensure the quality of the dataset by only using the cases with high inter-annotator agreement (line 1253).
>
> > What do you mean by "ecologically valid"? Do you mean naturally occurring negative examples?
>
> Yes, we are referring to naturally occurring negative examples which are not artificially created. We will clarify our usage of the term.
>
> > "In total, only 8.6% of the claims included one of the two types of errors." ... What are the two types?
>
> It refers to the Completeness and Correctness errors, which are described in line 142.
>
> > "we re-retrieve the cited web article(s)" ... Why did you need to re-retrieve them?
>
> We re-retrieved the cited web articles because SIDE only contains one evidence web article even when there are multiple on the original Wikipedia page (line 1166).
>
> > "Also, we filter claims that are decomposed into either only one or more than six subclaims." ... Why did you need this filter? Why six?
>
> First, we removed the claims that cannot be decomposed by the Claim-Split because we intend to focus on complex claims. Second, we removed the claims that are decomposed into more than six sub-claims because typically they are not interesting cases; they often just include a list of examples. Six was a threshold where we observed that the examples became very unnatural.
>
> > In Table 3, WiCE Subcl and WiCE Claim do not add up to 100%.
>
> We annotated 50 cases but claims/subclaims that are annotated as non-supported are excluded from this table because there is no entailment category for them. Therefore, the percentage for each category is not always an integer and does not necessarily add up to 100%. We will explain this clearly in the final version of this paper.
>
> > "p(SUPPORTED) + 0.5 × p(PARTIALLY-SUPPORTED)" ... How was this derived?
>
> The intuition behind this formula is that we want to prefer the sentences that receive high scores for the "supported" category, while also accepting the partially supporting sentences. The parameter 0.5 was arbitrarily selected. We will explain this point in the final version of the paper.
>
> > "As there can be multiple gold sets of supporting sentences for each claim/subclaim in WICE," ... I could not understand this clause.
>
> In the data collection process of WiCE, we retain individual sets of supporting sentences by all workers who chose the majority entailment label (line 1239).
>
> For example, when annotators provide the following annotations,
> * Annotator 1: entailment=supported, sentences=[1, 5, 9]
> * Annotator 2: entailment=supported, sentences=[1, 5, 18]
> * Annotator 3: entailment=supported, sentences=[1, 5, 9]
> * Annotator 4: entailment=supported, sentences=[1, 3, 9]
> * Annotator 5: entailment=partially_supported, sentences=[1, 5]
>
> The aggregated annotation for this sub-claim will be entailment=supported, supporting sentences={[1, 5, 9], [1, 5, 18], [1, 3, 9]}
>
> We use this strategy because there can be multiple different sets of sentences that correctly support the claim. In this example, it is possible that sentences [1, 5, 9], [1, 5, 18], and [1, 3, 9] all include sufficient information, but taking the union of these would be redundant. We will improve the description in the final version of our paper.
>
> > In Table 6, why was ANLI+WiCE worse than WiCE in the "w/ evidence context" setting, while it was better in the other setting?
>
> In this experiment, we pre-trained the model on ANLI, which does not have the evidence context, and further trained it on WiCE, which has the evidence context. This mismatch may be a reason for the drop in performance.
>
> > I could not understand footnote 7.
>
> In the oracle setting, our goal is to emulate a scenario where we have an ideal retrieval model and, subsequently, assess the maximum potential performance on the entailment classification task. However, if we simply provide the gold supporting sentences, it can introduce undesirable biases. For example, if there are many supporting sentences, the model can infer that the claim is supported without reading them.

---

### Meta-Review · Area_Chair_z11D · 2023-09-19

**Recommendation:** 4

**Metareview:**

All the reviewers agree on the fact that the paper provides a valuable contribution to the domain. In particular, the proposed entailment dataset extracted from Wikipedia will be useful for textual entailment recognition research, to complement existing resources. The experiments are solid, and provide interesting insights for research on RTE.
The authors have addressed the reviewers' concern in the rebuttal phase (in particular the fact of clarifying the contributions of the paper, as well as some technical aspects and justification of choices in the experimental setting), providing detailed descriptions of the unclear aspects in the paper.

---

### Decision · Program_Chairs · 2023-10-07

**Decision:**

Accept-Main

**Comment:**

All the reviewers agree on the fact that the paper provides a valuable contribution to the domain. In particular, the proposed entailment dataset extracted from Wikipedia will be useful for textual entailment recognition research, to complement existing resources. The experiments are solid, and provide interesting insights for research on RTE.
The authors have addressed the reviewers' concern in the rebuttal phase (in particular the fact of clarifying the contributions of the paper, as well as some technical aspects and justification of choices in the experimental setting), providing detailed descriptions of the unclear aspects in the paper.